# Total Soil CO_2_ Efflux from Drained Terric Histosols

**DOI:** 10.3390/plants13010139

**Published:** 2024-01-04

**Authors:** Egidijus Vigricas, Dovilė Čiuldienė, Kęstutis Armolaitis, Kristine Valujeva, Raija Laiho, Jyrki Jauhiainen, Thomas Schindler, Arta Bārdule, Andis Lazdiņš, Aldis Butlers, Vaiva Kazanavičiūtė, Olgirda Belova, Muhammad Kamil-Sardar, Kaido Soosaar

**Affiliations:** 1Institute of Forestry, Lithuanian Research Centre for Agriculture and Forestry, Liepų Str. 1, Girionys, LT-53101 Kaunas, Lithuania; egidijus.vigricas@lammc.lt (E.V.); kestutis.armolaitis@lammc.lt (K.A.); vaiva.kazanaviciute@lammc.lt (V.K.); olgirda.belova@lammc.lt (O.B.); 2Scientific Laboratory of Forest and Water Resources, Latvia University of Life Sciences and Technologies, Liela Str. 2, LV-3001 Jelgava, Latvia; kristine.valujeva@lbtu.lv; 3Natural Resources Institute Finland (Luke), Latokartanonkaari 9, FI-00790 Helsinki, Finland; raija.laiho@luke.fi (R.L.); jyrki.jauhiainen@luke.fi (J.J.); 4Department of Geography, University of Tartu, Vanemuise 46, 51003 Tartu, Estonia; thomas.schindler@ut.ee (T.S.); kamil.sardar@ut.ee (M.K.-S.); kaido.soosaar@ut.ee (K.S.); 5Latvian State Forest Research Institute ‘Silava’ (LSFRI Silava), Rigas Str. 111, LV-2169 Salaspils, Latviaandis.lazdins@silava.lv (A.L.); aldis.butlers@silava.lv (A.B.)

**Keywords:** total soil CO_2_ efflux, drained peatland, Terric Histosols, perennial grassland, Norway spruce, black alder, silver birch

## Abstract

Histosols cover about 8–10% of Lithuania’s territory and most of this area is covered with nutrient-rich organic soils (Terric Histosols). Greenhouse gas (GHG) emissions from drained Histosols contribute more than 25% of emissions from the Land Use, Land Use Change and Forestry (LULUCF) sector. In this study, as the first step of examining the carbon dioxide (CO_2_) fluxes in these soils, total soil CO_2_ efflux and several environmental parameters (temperature of air and topsoil, soil chemical composition, soil moisture, and water table level) were measured in drained Terric Histosols under three native forest stands and perennial grasslands in the growing seasons of 2020 and 2021. The drained nutrient-rich organic soils differed in terms of concentrations of soil organic carbon and total nitrogen, as well as soil organic carbon and total nitrogen ratio. The highest rate of total soil CO_2_ efflux was found in the summer months. Overall, the rate was statistically significant and strongly correlated only with soil and air temperature. A trend emerged that total soil CO_2_ efflux was 30% higher in perennial grassland than in forested land. Additional work is still needed to estimate the net CO_2_ balance of these soils.

## 1. Introduction

Globally, peat soils (Histosols) cover approximately 3% of the land area and have accumulated about one third of total soil organic carbon stocks [1,2,3]. One third of all organic soils are found in Europe [4]. Peatlands have an extremely important role in the global carbon cycle. Globally, the European Union (EU) is the second largest source of greenhouse gas (GHG) emissions from drained peatlands, with 220 Mt CO_2_-C annual emissions from peatlands. Lithuania is recognized as one of 11 EU countries in which GHG emissions from drained Histosols contribute more than 25% of Land Use, Land Use Change and Forestry (LULUCF) emissions [5].

In Lithuania, Histosols cover about 8–10% of the terrestrial territory, and the largest part, 513 kha (about 78%) [6,7,8], is occupied by nutrient-rich organic soils classified as Terric [9] or Sapric [10] Histosols. Most of these soils, approximately 94%, have been drained [8]. About half of all drained Terric Histosols are forested, and the remaining half are used for croplands and perennial grasslands [8]. Drained and forested Terric Histosols are usually dominated by tree species such as silver birch (*Betula pendula* Roth), black alder (*Alnus glutinosa* (L.) Gaertn.), and Norway spruce (*Picea abies* (L.) H. Karst.) [11,12].

Like other EU member states, Lithuania is obligated by the Kyoto Protocol to report annual GHG emissions from LULUCF in its national GHG inventory report. Currently, the State Forest Service uses default CO_2_-C annual emission factors (EF) to represent drained peatlands when reporting GHG inventories in Lithuania [13]. These Tier 1 EFs are determined by using relatively wide categories for averaging annual soil carbon balance values from peer-reviewed studies performed in boreal and cold temperate regions [14]. More recently, the CO_2_-C EFs were significantly increased from 0.68 to 2.6 t C ha^−1^ y^−1^ for forest land and from 0.25 to 6.1 t C ha^−1^ y^−1^ for fertile grasslands on drained organic soils in temperate regions [15].

Total soil CO_2_ efflux is the second largest carbon flux (after photosynthesis) in terrestrial ecosystems [16] and is a result of autotrophic (CO_2_ produced by vegetation living roots) and heterotrophic (CO_2_ produced by soil microbes in the decomposition of organic matter deposited into soil and on the soil surface) soil respiration [17,18,19]. The total soil respiration rate depends on many variables such as local climate, land use, nutrient availability, soil temperature, soil moisture, and water table level (WTL) [20,21,22,23]. Generally, lower WTL in drained areas has a positive effect on oxygen availability, which can enhance oxidation and decomposition at greater peat layer depth from the soil surface [24,25]. Furthermore, soil CO_2_ efflux strongly depends on soil organic matter (SOM) decomposition due to increased biomicrobial biomass and activity [26,27]. It is well documented that soil organic carbon (SOC), total nitrogen (TN), and the soil organic carbon/total nitrogen (C/N) ratio are the main indicators of SOM quality [28,29], and these parameters usually differ between forest and grassland ecosystems [30].

Despite numerous studies investigating autotrophic and heterotrophic soil CO_2_ efflux from drained Histosols, studies on soil CO_2_ emission trends in hemi-boreal regions are still scarce and form an urgent need for data applicable to regional or local conditions [4].

In this study, we monitored total soil CO_2_ efflux and environmental factors such as air temperature, soil physical parameters (temperature and moisture of the topsoil), soil chemical parameters (SOC, TN, and C/N in the topsoil), and soil water level table in three forest types and perennial grassland. These types of land cover are commonly found in drained nutrient-rich organic soils, more specifically Terric Histosols, in Lithuania. The objectives were: (1) to determine the concentrations of SOC and TN in the topsoil of drained Terric Histosols under stands of different tree species and perennial grassland; (2) to investigate the variation in total soil CO_2_ efflux across different months and between stands of different tree species and perennial grassland; and (3) to assess the environmental factors that may have the greatest influence on total soil CO_2_ efflux in drained Terric Histosols under perennial grassland and forest stands. The hypothesis of this study is that the concentrations of SOC and TN, along with the C/N ratio, are the key factors influencing total soil CO_2_ efflux in forest and grassland biotopes.

## 2. Results

### 2.1. Concentrations of Soil Organic Carbon and Total Nitrogen in Drained Terric Histosols

The variation between the study sites in the concentrations of soil organic carbon (SOC) and total nitrogen (TN) as well as the SOC/TN (C/N) ratio in the drained Terric Histosols was statistically significant (Figure 1).

The highest mean concentration of SOC (46%), highest C/N ratio (25), and the lowest concentration of TN (1.7%) were in the topsoil of the Norway spruce stand (Figure 1). A high mean concentration of SOC (43%) was also found in the topsoil of the silver birch stand, while the C/N ratio was intermediate (19). The concentration of TN in all study sites, except the spruce stand, was similar and ranged between 2.0% and 2.3%. Significantly lower concentrations of SOC (33% in perennial grassland and 31.6% in the black alder stand) and lower C/N ratios (14.4 in grassland and 16.8 in the black alder stand) were observed in the topsoil of the perennial grassland and the black alder stand. We interpreted the relatively high concentration of SOC and high C/N ratio as indicating a lower decomposition rate of peat in the silver birch and, especially, the Norway spruce stand, while the opposite results in black alder and perennial grassland were taken to indicate a high peat decomposition rate and high nitrogen release during the decay process [31].

### 2.2. Total Soil CO_2_ Efflux under Forest and Grassland Sites

In 2021, total soil CO_2_ efflux in the grassland site and the three forest sites varied from 0.12 g CO_2_ m^−2^ h^−1^ to 1.5 g CO_2_ m^−2^ h^−1^ (Figure 2). The lowest values for total soil CO_2_ efflux (0.12–0.45 g CO_2_ m^−2^ h^−1^) occurred in the spring and in the autumn. Higher values of mean total soil CO_2_ efflux were detected in the summer, ranging from 0.5 g CO_2_ m^−2^ h^−1^ to 1.5 g CO_2_ m^−2^ h^−1^. Furthermore, in summer the differences between the sites became apparent and the highest total soil CO_2_ efflux was observed in the perennial grassland site, ranging on average from 0.8 g CO_2_ m^−2^ h^−1^ to 1.5 g CO_2_ m^−2^ h^−1^, while the effluxes in the forest sites were lower with averages of between 0.5 and 0.9 g CO_2_ m^−2^ h^−1^.

The results obtained in 2022 indicated that total soil CO_2_ efflux increased at the beginning of the vegetative period, in a similar pattern as was found in 2021 (Figure 2). In spring (April and May), total soil CO_2_ efflux varied on average between 0.37 and 0.51 g CO_2_ m^−2^ h^−1^ in the perennial grassland and in the silver birch stand, and on average between 0.27 and 0.29 g CO_2_ m^−2^ h^−1^ in the Norway spruce stand. In June, total soil CO_2_ efflux increased on average by 40% in the Norway spruce stand, from 40% to 50% in the perennial grassland, and from 40% to 55% in the silver birch stand, as compared with the total soil CO_2_ efflux values found in May and April, respectively. Furthermore, the highest total soil CO_2_ efflux (an average of 0.8 g CO_2_ m^−2^ h^−1^) was determined in the black alder stand in June, possibly for the reasons discussed in Section 4.2.

### 2.3. The Influence of Environmental Factors on Total Soil CO_2_ Efflux

The variations of environmental factors such as air and soil temperature, soil moisture, and WTL are presented in Table 1.

The data obtained in our study led us to evaluate the effect of environmental factors on total soil CO_2_ efflux (Figure 3, Table 2). The studied environmental factors such as soil chemical properties (SOC, TN, and the C/N ratio in the topsoil), WTL, soil and air temperature (in the topsoil), different forms of land use (forest and grassland sites), and the depth of drainage ditches were included in principal components analysis (PCA) (Figure 3).

PCA indicated a strong positive relationship between air and soil temperatures and total soil CO_2_ efflux (Figure 3, Component 2). The efflux further showed a negative relationship to wetness (variables WTL and moisture). The depth of drainage ditches was associated with the SOC concentration and C/N ratio in the topsoil (Figure 3, Component 1) of drained Terric Histosols. TN was not included in any of the groups, because TN concentrations were similar in all studied forest and grassland sites (Figure 1).

The correlation analysis also indicated that the strongest significant and positive linear relation (*r* = 0.7, *p* < 0.05) was found between air and soil temperature and total soil CO_2_ efflux (Table 2). Total soil CO_2_ efflux strongly depended on air and soil (at 10 cm depth) temperatures and total soil CO_2_ efflux tended to increase with increasing air and soil temperatures (Table 1; Figure 2).

Soil moisture (*r* = −0.4, *p* < 0.05) and WTL (r = −0.3, *p* < 0.05) were moderately and negatively correlated with total soil CO_2_ efflux, indicating that increasing moisture and higher WTL resulted in lower efflux. Meanwhile, a weak positive relation (*r* = −0.2, *p* > 0.05) was observed between total nitrogen (TN) concentration in the topsoil and total soil CO_2_ efflux and, conversely, a weak negative (*r* = 0.2, *p* > 0.05) relation was found between SOC and SOC/TN (C/N) ratio.

## 3. Discussion

Numerous studies investigating soil CO_2_ efflux from drained peatlands have been conducted in Fennoscandia and in the Baltic countries. This may be due to the fact that peatlands cover 31–34% of Sweden and Finland, around 20% of Estonia, around 10% of Latvia and Lithuania, and about 7.7% of Norway’s territory [32,33,34,35]. Several studies in which total soil CO_2_ efflux measurements were performed using a closed chamber system were selected for comparison (Table 3). A study conducted in the boreal region of Russia reported that the total soil CO_2_ efflux released from Histosols in black alder forests could be approximately 0.33 g CO_2_ m^−2^ h^−1^ [36]. Another study performed in Finland reported that the annual total CO_2_ efflux in less fertile forests was an average of 2000 g CO_2_ m^−2^ y^−1^, and an average of 3000 g CO_2_ m^−2^ y^−1^ in fertile forests [37]. Meanwhile, total soil CO_2_ effluxes in perennial grasslands and agricultural croplands have been found to have similar values [38], with annual emissions were approximately 5–7 t CO_2_ ha^−1^ y^−1^ [39]. The total soil CO_2_ efflux in grasslands can be about two times higher during the growing season in the hemi-boreal zone than in the boreal zone (Table 3). In this study, the highest total soil CO_2_ efflux released from perennial grassland in summer (average of 1.16 g CO_2_ m^−2^ h^−1^) is well in line with a recent study evaluating total soil CO_2_ efflux in Histosols under grasslands in Sweden and Latvia [40,41]. Also, similar results of total soil CO_2_ efflux from forest sites were obtained in the study conducted in Latvia [41]. Studies investigating annual CO_2_ emissions have reported that total soil CO_2_ efflux in winter ranged up to 5–30% of the total CO_2_ efflux measured during the growing season [39,42].

Oertel et al. [23] summarised studies conducted in the last two decades and reported that local climate conditions, land use and land cover, soil temperature and humidity, and soil fertility are the key drivers for GHG emissions from various types of soils. In general terms, the findings of the present study reveal that forest and grassland biotopes differ by SOC, TN concentrations, and C/N ratio in the topsoil of drained Terric Histosols. However, SOC, TN, and C/N ratios did not exhibit the expected relationship with total soil CO_2_ efflux (Table 2). In addition, the mean total soil CO_2_ efflux calculated over the entire vegetation period during the study did not show significant (*p* > 0.05) differences among the Norway spruce, black alder, and silver birch stands and perennial grassland (Figure 4). The obtained data only show that total soil CO_2_ efflux was on average 30% higher in grassland than in forested Terric Histosols.

The higher total soil CO_2_ efflux found in perennial grassland could be related to a huge alteration in the local mix of herbaceous species during the growing season (i.e., when one species dies, others grow). Other studies have reported that higher total soil CO_2_ efflux in perennial grasslands may be related to the higher fine root biomass (to which autotrophic soil CO_2_ efflux is attributed) in grasslands compared to forest ecosystems [43,44].

Furthermore, the Norway spruce, black alder, and silver birch forest stands differed from each other in terms of both dendrometric parameters (Appendix A) and the species composition of their understory vegetation cover (see Section 4.1). Such differences could have an impact on the amount of solar radiation entering the studied forest stands and their understory vegetation, and could, therefore, accelerate or reduce autotrophic total soil CO_2_ efflux.

The soil WTL fluctuated across the entire study period (Table 1), which could be related to the relative depth of the drainage systems (using channelized forest streams) present in the studied forest and perennial grassland sites (see Section 4.1). Also, precipitation distribution over time and rain volumes could have an impact on the soil water table level. It is important to note that WTL was at least 40 cm below the soil surface at all studied sites (Table 1). The topsoil of the studied drained Terric Histosols had good aeration conditions. This indicates that neither the relative depth of drainage systems nor fluctuations in the WTL had significant effects on total CO_2_ efflux (Figure 3; Table 1).

In addition, since total soil CO_2_ efflux is the sum of several processes taking place in the soil, including heterotrophic respiration from the decomposition of organic matter, autotrophic respiration by live roots, and possibly oxidation of methane produced in deep anoxic soil layers, it is challenging to interpret its variation when detailed results on the actual constituent processes are not available. On the other hand, the results obtained by this study confirm the findings of previous studies [45,46,47] and indicate that fluctuations in soil and air temperatures are the most important factors determining the seasonal (spring, summer, and autumn) variations of total soil CO_2_ efflux. Several studies have reported that higher soil temperature has a positive influence on the activity of soil microbiota and promotes CO_2_ production by both autotrophic and heterotrophic soil respiration [48,49].

## 4. Material and Methods

### 4.1. Study Sites

This study was performed in the southern part of Lithuania (Figure 5). The study area belongs to the hemi-boreal forest zone, with an annual average temperature of 6.5 °C and mean annual rainfall of 650 mm [50]. The study sites (four in total) were located on drained nutrient-rich organic soils classified as Sapric [10] or Terric [9] Histosols, in Norway spruce (*Picea abies* (L.) H. Karst.), black alder (*Alnus glutinosa* (L.) Gaertn.), and silver birch (*Betula pendula* Roth) stands, and in a perennial grassland (Figure 5, Appendix A).

Forests containing the three studied tree species, along with perennial grassland, represent 60% of all drained Terric Histosols found in Lithuania [8,53]. All four study sites were drained using channelized forest streams (existing forest streams that were straightened, widened, and deepened) for the first time in 1958. The peat layer was 50 cm thick in the perennial grassland site and >100 cm thick in all three studied forest stands.

The first study site was in a pure 70-year-old Norway spruce stand. This site was drained for forestry purposes using channelized forest streams of 1.6 m depth. Compared with the other studied forest stands, the Norway spruce stand was the oldest and had the highest total stem volume and height (Appendix A). The ground vegetation was dominated by moss red-stemmed feathermoss (*Pleurozium schreberi* (Brid.) Mitt.) and dwarf shrub blueberry (*Vaccinium myrtillus* L.).

The second study site was in a 43-year-old silver birch forest stand with black alder admixture. This site was drained using a channelized forest stream of 1.2 m depth. The ditch was cleaned by removing a layer of organic sediment in 2015. Nevertheless, the studied stand had the lowest stem volume of the three forest sites (Appendix A). The ground vegetation was dominated by herbs such as wood-sorrel (*Oxalis acetosella* L.), northern beech fern (*Phegopteris connectilis* (Michx.) Watt), and may lily (*Maianthemum bifolium* (L.) F.W. Schmidt).

The third study site was in a pure 30-year-old planted black alder stand. This stand was drained using channelized forest streams of 0.6 m depth. The black alder stand had the lowest DBH and, conversely, the highest tree density (Appendix A). The ground vegetation was dominated by raspberry (*Rubus idaeus* L.) shrubs.

The fourth study site was in a drained perennial grassland area. The site was drained using a channelized forest stream of 0.6 m depth, and covered with grass species such as milk-parsley (*Peucedanum palustre* (L.) Moench), meadowsweet (*Filipendula ulmaria* (L.) Maxim.), common marsh bedstraw (*Galium palustre* L.), common couch (*Elymus repens* (L.) Gould), wild mint (*Mentha arvensis* L.), common nettle (*Urtica dioica* L.), and creeping bent grass (*Agrostis stolonifera* L.).

### 4.2. Measurements of Total Soil CO_2_ Efflux and Environmental Factors

Measurements were carried out once per month from the beginning of May to the end of October 2021 and from April to June 2022. Three subplots for total soil CO_2_ efflux measurements were established in each study site. The distance between subplots was 30 m. Total soil CO_2_ efflux was measured in each subplot, and with at least three replicates, using a closed chamber system that includes a portable CO_2_ analyzer (EGM-4, PP systems, Amesbury, MA, USA) with a chamber (diameter 32 cm; height 18 cm). The chamber was installed on the sample plot directly over a 2 cm deep groove in the ground. The CO_2_ concentration was measured for a 120 s period. Total soil CO_2_ was measured only during the daytime when there was no precipitation, from 11 am to 2 pm. No data were collected during the cold period due to snow or ice cover on the soil. It was also impossible to measure the total soil CO_2_ efflux in the black alder stand in April and May 2022 because the study area was destroyed by wild boars in spring, and soil CO_2_ efflux values in disturbed peat can be up to three times higher than in undisturbed peat [54].

In total, 324 measurements were performed during the entire study period. CO_2_ efflux (g CO_2_ m^−2^ h^−1^) was calculated from the linear change of CO_2_ concentration (ppm) in time as a function of chamber volume, air temperature, and air pressure [55]. Also, the rate was calculated from the linear change (*R*^2^ ≥ 0.95) in CO_2_ concentration during the measurement period.

Alongside the CO_2_ efflux measurements, soil and air temperature, soil moisture, and soil water table level were simultaneously measured in each subplot of each study site (Table 1). Air temperature was measured in the shade using a thermometer (Comet system, Rožnov pod Radhoštěm, Czech Republic), soil temperature (at a depth of 0–10 cm) was measured using soil temperature probe (Comet system, Czech Republic), soil water table level measured using a water level probe (Odyssey^®^, Christchurch, New Zealand), and soil moisture was determined (at a depth of 0–10 cm) using a soil moisture meter (HH2, Delta-T Devices, Burwell, UK).

### 4.3. Soil Sampling and Chemical Analysis

Soil sampling was conducted in October 2021 after the growing season. Composite soil samples were taken from three systematically distributed points 28 m apart from each other. Three square-shaped 40 cm × 40 cm pits were dug at each sampling point. Samples of topsoil (0–10 cm depth) were taken from all sides of the pit in three replicates using a 100 cm^3^ cylinder. In total, 108 topsoil samples were taken for chemical analysis at the Agrochemical Research Laboratory of the Lithuanian Research Centre for Agriculture and Forestry. Soil organic carbon (SOC) [56] and total nitrogen (TN) [57] were determined by dry combustion at 900 °C with a CNS analyzer (Elementar Analy-sensysteme GmbH, Langenselbold, Germany).

### 4.4. Statistical Analyses

The normality of the data was checked with Kolmogorov–Smirnov and Shapiro–Wilk tests, and all the data were found to be normally distributed. The impact of different forms of land use and different months, both treated as categorical variables, on soil CO_2_ efflux was determined using univariate analysis of variance with Tukey’s HSD test with significance level α = 0.05. Univariate analysis of variance with Tukey’s HSD test was also used to compare SOC, TN, and the C/N ratio between the study sites. Pearson correlation analysis was used as a measure of the strength of the linear relation between CO_2_ efflux and abiotic factors such as soil moisture, soil water table level (WTL), soil temperature at 10 cm depth (T soil 10 cm), and air temperature (T air).

Principal component analysis (PCA) was used to determine whether there were any significant interactions between total soil CO_2_ efflux and several environmental factors, including soil physical parameters (soil moisture and soil temperature at 10 cm depth), soil chemical composition (SOC, TN, and C/N ratio at 10 cm depth), air temperature, WTL, and the depth of drainage ditches. PCA was performed using Varimax rotation. All statistical analyses were conducted using IBM SPSS 22.0 (IBM, New York, NY, USA).

## 5. Conclusions

This study was performed in drained Terric Histosols under Norway spruce, silver birch, and black alder stands and perennial grassland with the aim of measuring total soil CO_2_ efflux. The nutrient-rich organic soils under the Norway spruce and silver birch stands were distinguished by significantly higher C/N ratios and concentrations of SOC and TN, indicating lower peat decomposition rates. In contrast, the opposite results found in the soils under the black alder stand and perennial grassland were interpreted to indicate a high rate of peat decomposition and nitrogen release during the peat decay process. However, the hypothesis that SOC, TN, and C/N ratio are the key factors influencing total soil CO_2_ efflux from drained Terric Histosols under forest and grassland biotopes was not supported. The results indicate a trend that total CO_2_ efflux was 30% higher in drained Terric Histosols under perennial grassland than it was under forest stands. However, air and topsoil temperatures were the most important of all studied environmental parameters (soil chemistry, land use, WTL, and soil moisture), explaining the variation of the total CO_2_ efflux from drained Terric Histosols. In order to form national emission factors, it may be necessary to carry out flux monitoring throughout the year (also in winter). Based on soil total CO_2_ efflux measurements, from a climate change mitigation perspective, forest stands are a better land-use option for reducing CO_2_ emissions than perennial grasslands. However, quantifying the full net CO_2_ balance of these soils requires further measurements.

## Figures and Tables

**Figure 1 plants-13-00139-f001:**
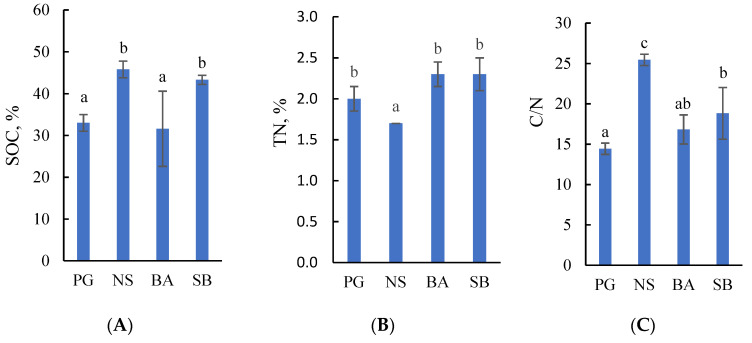
Mean concentrations of soil organic carbon (SOC) (**A**), total nitrogen (TN) (**B**), and SOC/TN (C/N) ratio (**C**) in the topsoil (0–10 cm) of perennial grassland (PG) and Norway spruce (NS), black alder (BA), and silver birch (SB) stands. The results are expressed as mean ± SE (*n* = 108). The different lower-case letters indicate statistically significant differences between the sites, *p* ˂ 0.05 (Tukey’s HSD test).

**Figure 2 plants-13-00139-f002:**
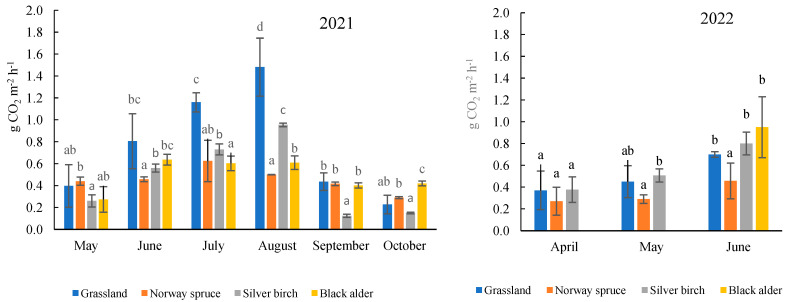
The distribution of the mean monthly total soil CO_2_ efflux in perennial grassland and Norway spruce, silver birch, and black alder stands during the vegetative period (2021–2022). The data represent mean ± SE (*n* = 324). The different lower-case letters indicate statistically significant differences between the sites, *p* ˂ 0.05 (Tukey’s HSD test).

**Figure 3 plants-13-00139-f003:**
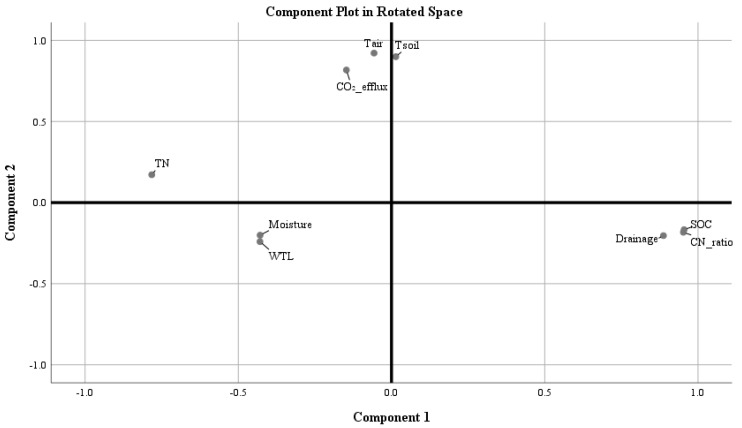
PCA of soil chemical properties (SOC, TN, and the C/N ratio in the topsoil), water table level, soil and air temperature (in the topsoil), drainage (the depth of drainage ditches), and total soil CO_2_ efflux. *The x*-axis represents principal component 1, and the *y*-axis component 2. Abbreviations: CO_2__efflux—soil total CO_2_ efflux; Tair—air temperature; Tsoil—soil temperature at 10 cm depth; WTL—water table level; Moisture—soil moisture at 5 cm depth, SOC—soil organic carbon determined at 0–10 cm soil depth soil; TN—total nitrogen determined at 0–10 cm soil depth; CN_ratio—SOC/TN ratio determined at 0–10 cm soil depth; Drainage—depth of the drainage ditches.

**Figure 4 plants-13-00139-f004:**
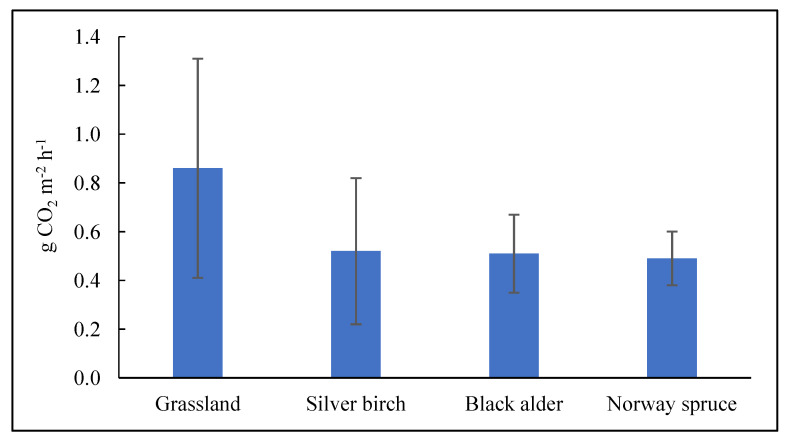
Distribution of mean total soil CO_2_ efflux over the entire study period (from May 2021 to June 2022) among the three studied forest sites and the perennial grassland site. The data represent mean ± SE.

**Figure 5 plants-13-00139-f005:**
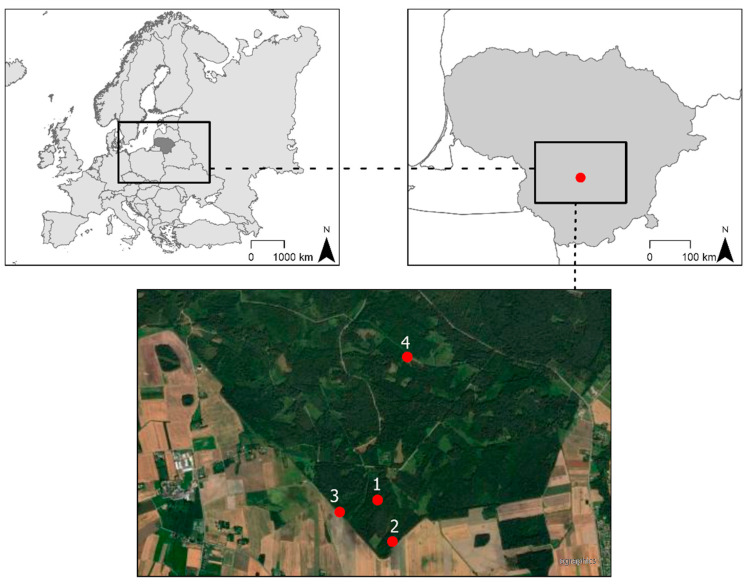
Study sites: 1—Norway spruce stand (54°47′37.5″ N 24°04’50.6″ E); 2—silver birch stand (54°47′23.4″ N 24°04′59.3″ E); 3—black alder stand (54°47′33.4″ N 24°04′28.2″ E); 4—perennial grassland (54°48′26.0″ N 24°05′08.1″ E) [51,52].

**Table 1 plants-13-00139-t001:** Seasonal variation of air and soil temperatures, soil moisture, and soil water table level (WTL) in the studied Norway spruce, silver birch, and black alder stands and perennial grassland.

	2021			2022	
	Spring ^a^	Summer ^b^	Autumn ^c^	Spring ^d^	Summer ^e^
	Norway spruce				
Air temperature, °C	17 ± 0	26 ± 2	12 ± 1	14 ± 3	15 ± 0
Soil temperature, °C	11 ± 1.0	19 ± 1	10 ± 1	10 ± 2	14 ± 0
Soil moisture, %	20 ± 5	16 ± 4	19 ± 5	20 ± 7	22 ± 2
WTL, cm	−40 ± 4	−91 ± 9	−95 ± 9	−67 ± 9	−42 ± 15
	Silver birch				
Air temperature, °C	19 ± 0	26 ± 4	16 ± 1	13 ± 9	25 ± 0
Soil temperature, °C	12.6 ± 0.2	17.3 ± 2.5	11.0 ± 2.8	6 ± 6	15 ± 4
Soil moisture, %	22 ± 3	19 ± 5	20 ± 6	28 ± 1	16 ± 4
WTL, cm	−100 ± 0	−140 ± 0	−130 ± 0	−130 ± 0	−88 ± 1
	Black alder				
Air temperature, °C	15 ± 0	27 ± 3	17 ± 2	15 ± 8	25 ± 0
Soil temperature, °C	13 ± 0	19 ± 2	13 ± 4	10 ± 1	14 ± 0
Soil moisture, %	29 ± 5	24 ± 1	23 ± 1	37 ± 12	30 ± 1
WTL, cm	−40 ± 15	−87 ± 32	−101 ± 14	−47 ± 9	−30 ± 7
	Perennial grassland				
Air temperature, °C	19 ± 0	29 ± 4	18 ± 6	15 ± 2	19 ± 0
Soil temperature, °C	7 ± 1	18 ± 3	13 ± 5	6 ± 2	10 ± 2
Soil moisture, %	80 ± 35	32 ± 15	64 ± 21	82 ± 20	100 ± 0
WTL, cm	−52 ± 10	−67 ± 20	−46 ± 19	−28 ± 11	0 ± 0

Notes: The data represent mean ± SE. Soil temperature and soil moisture were measured in the topsoil (10 cm depth). The data of environmental variables was collected in: ^a^ May 2021; ^b^ June, July, and August 2021; ^c^ September and October 2021; ^d^ April and May 2022; ^e^ June 2022.

**Table 2 plants-13-00139-t002:** Pearson’s correlation matrix showing relationships between environmental variables and total soil CO_2_ efflux in drained Terric Histosols in Norway spruce, silver birch, and black alder forest stands and in perennial grassland (*n* = 153).

	CO_2_ Efflux	T Air	T Soil	WTL	Moisture	SOC	TN	C/N
CO_2_ efflux	1.0	**0.7** *	**0.7** *	−0.3 *	−0.4 *	−0.2	0.2	−0.2
T air		1.0	**0.8** *	−0.1	−0.1	−0.2 *	0.2 *	−0.2 *
T soil			1.0	−0.1 *	−0.1	−0.2 *	0.1	−0.2 *
WTL				1.0	0.4 *	−0.3 *	0.2	−0.2 *
Moisture					1.0	−0.4*	0.2 *	−0.2 *
SOC						1.0	**−0.7** *	**0.9** *
TN							1.0	**0.9**
C/N								1.0

Notes: * Pearson’s correlation coefficients (*r*) with *p* < 0.05 are considered statistically significant. Abbreviations: T air—air temperature; T soil—soil temperature at 10 cm depth; WTL—water table level; SOC—concentration of soil organic carbon (%) at 0–10 cm depth; TN—concentration of total nitrogen (%) in soil at 0–10 cm depth; C/N—the ratio of SOC/TN.

**Table 3 plants-13-00139-t003:** Total soil CO_2_ efflux released from drained Histosols in boreal and hemi-boreal regions.

Forest Zone	Country	Vegetation Type	Study Period	Soil CO_2_ Efflux	Reference
Boreal	Finland	Grassland	Growing	4.84–7.00 t CO_2_ ha^−1^ y^−1^ (from ~0.50 to ~1.00 g CO_2_ m^−2^ h^−1^)	[39]
Boreal	Finland	Grassland	Winter	130–150 kg CO_2_ ha^−1^ y^−1^ (from ~0.00 to ~0.40 g CO_2_ m^−2^ h^−1^)	
Boreal	Finland	Grassland	Growing	0.35 g CO_2_ m^−2^ h^−1^	[38]
Boreal	Finland	Cropland	Growing	0.33 g CO_2_ m^−2^ h^−1^	
Boreal	Russia	Forest	Growing	0.33 g CO_2_ m^−2^ h^−1^	[36]
Boreal	Finland	Forest	Annual	2000 g CO_2_ m^−2^ y^−1^ (0.23 g CO_2_ m^−2^ h^−1^)	[37]
Boreal	Finland	Forest	Annual	3000 g CO_2_ m^−2^ y^−1^ (0.34 g CO_2_ m^−2^ h^−1^)	
Hemi-boreal	Sweden	Grassland	Growing	0.74 g CO_2_ m^−2^ h^−1^	[40]
Hemi-boreal	Sweden	Grassland	Spring	0.90 g CO_2_ m^−2^ h^−1^	
Hemi-boreal	Sweden	Grassland	Summer	1.12 g CO_2_ m^−2^ h^−1^	
Hemi-boreal	Sweden	Grassland	Autumn	0.40 g CO_2_ m^−2^ h^−1^	
Hemi-boreal	Latvia	Grassland	May-August	0.35 g CO_2_-C m^−2^ h^−1^ (1.28 g CO_2_ m^−2^ h^−1^)	[41]
Hemi-boreal	Latvia	Forest	May-August	0.23 CO_2_-C m^−2^ h^−1^ (0.84 g CO_2_ m^−2^ h^−1^)	

## Data Availability

Data are contained within the article and Appendix A.

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
