# Peer review of "Total Soil CO2 Efflux from Drained Terric Histosols"

_plants, 2024, doi:10.3390/plants13010139_

Round 1
Reviewer 1 Report
Comments and Suggestions for Authors
The authors conducted CO2 efflux measurements on the top soils of Terric Histosols, which were covered with various vegetation types. They also aimed to identify correlations between these rates and soil parameters. Overall, the study demonstrates considerable merit. However, I have a few questions regarding the methodologies employed by the authors.
Comments :
Please review the consistency of font sizes across the document as some sections appear larger than others.
Consider enhancing Figure 3 and Table 1 to augment their informativeness. Utilizing heatmaps and employing alternative statistical methods could potentially enhance the visualization and representation of connections between parameters.
Regarding Method 4.1, it would be beneficial to explain the criteria for selecting the four sites. Are they sufficiently representative of the broader area under study?
Line 302, it's essential to clarify whether controls were implemented during the collection of CO2 efflux measurements to ensure accurate and reliable measurements.
Comments on the Quality of English LanguageNO.
Author Response
Dear reviewer,
Thank you for your time during this special period and comments. We took into account when correcting the manuscript. Sometime we explained our choices more deeply.

Reviewer 2 Report
Comments and Suggestions for Authors
General Coments
The study analyzed the carbon dioxide fluxes in a region with peat soils transformed to forest or grasslands over drained Histosoils u in the growing seasons of 2020 and 2021 and used as a proxy of this parameter temperature of air and topsoil, soil chemical composition, soil moisture, and water table level under three native forest stands and perennial grasslands in the growing seasons of 2020and 2021.
They found the highest CO2 efflux in the summer since is statistically strongly correlated with temperature. In addition, mentioned that the perennial grassland CO2 efflux was higher than in forested land.
The study is really interesting, nevertheless the number of simples seems insufficient.
The introduction is well-focused, going from the most general to the most specific, uses a lot of updated bibliography and addresses the gap in knowledge well. The objectives are well-defined and related to the hypothesis.
I don´t understand why the Material and methods are in the 4th part of this document. It is difficult to understand the abbreviations of the results since they are defined in a later section. It makes reading the manuscript difficult. I advise changing the order of this section and putting the M&M after the introduction as is customary in all scientific articles.
I think that only 3 subplots per ecosystem type is not enough to make a robust statistic. It is difficult to understand the deviation of each study site with only 3 subsamples
The statistics are a bit weak, I recommend the use of mixed models
I do believe that the C and N part is publishable, although with an improvement in statistics. However, I believe that the measurement of CO2 flow should increase the sample size and include data collection from targets.
It is not appropriate to carry out the discussion and conclusions since it is based on an erroneous statistic.
Minor comments
Abstract
L15: where say Beng must said is
L17: where say emission must said emisión
L26: where say needed for estimating must said needed to estimate
Introduction
L53-55: the letter size seems different.
Materials and methods
I would include Table 4 as an annex, and some of this information is already in the text. Decide if the information goes as a table or as text
Figure 5, I would mention here the abbreviations used in the results to name each study area,
Targets are not included in the measurement of CO2 flux, that is, a chamber without contact with the ground to measure the possible entry of CO2 from outside the chamber
Table 5 results should not go in this section
The PCA analysis includes the type of ecosystem (Pasture or forest) as a variable. The PCA analysis should only include soil data and ecosystem type vectors can then be projected. I recommend the Vegan R package with the enfit functions for this type of analysis
Results
Figure 3. Axis numbers must be separated by dots, not commas.
Discussion
Figure 4 should go in results
Author Response
Dear Reviewer,
Thank you for your time during this period and comments. We took into account when correcting the manuscript. Sometime we explained our choices more deeply.
Please find answers to your comments and new manuscript with supplementary material

Round 2
Reviewer 2 Report
Comments and Suggestions for Authors
Thank you for your comments . They have satisfied my doubts